# Composite Heat Sink Material for Superconducting Tape in Fault Current Limiter Applications

**DOI:** 10.3390/ma13081832

**Published:** 2020-04-13

**Authors:** Marcela Pekarčíková, Jozef Mišík, Marian Drienovský, Jozef Krajčovič, Michal Vojenčiak, Marek Búran, Marek Mošať, Tomáš Húlan, Michal Skarba, Eva Cuninková, Fedor Gömöry

**Affiliations:** 1Faculty of Materials Science and Technology in Trnava, Slovak University of Technology in Bratislava, Jána Bottu 2781/25, 917-24 Trnava, Slovakia; jozef.misik.tt@gmail.com (J.M.); marian.drienovsky@stuba.sk (M.D.); jozef_krajcovic@stuba.sk (J.K.); michal.skarba@stuba.sk (M.S.); eva.cuninkova@stuba.sk (E.C.); 2Institute of Electrical Engineering, Slovak Academy of Sciences, Dúbravska cesta 9, 841-04 Bratislava, Slovakia; elekvoje@savba.sk (M.V.); marek.buran@savba.sk (M.B.); marek.mosat@savba.sk (M.M.); elekgomo@savba.sk (F.G.); 3Faculty of Natural Sciences, Constantine the Philosopher University in Nitra, Tr. A. Hlinku 1, 949-74 Nitra, Slovakia; thulan@ukf.sk

**Keywords:** composite, thermal stabilization, high-temperature superconductor

## Abstract

We enhanced the performance of superconducting tapes during quenching by coating the tapes with various composites, with regards to the application of such coated systems in superconducting fault current limiters. In composition of the coating, we varied the type of epoxy matrix, the content of ceramic filler particles and the use of reinforcement in order to optimize the thermal and the mechanical stability of the coated tapes. By this way modified superconducting tapes were able to reduce the maximum temperature 170 °C of not modified superconducting tape to 55 °C during the quench with electric field up to 130 V m^−1^.

## 1. Introduction

High-temperature superconductors (HTSs) transport electric current without resistance. Nowadays, the HTSs are produced as a composite coated conductor (CC), where a REBCO (REBa_2_Cu_3_O_7-δ_ with RE = rare earth elements) superconducting layer is coated on metallic substrate and covered by silver and copper protection layers. They are available from several producers as a commercial product in form of approximately 0.1-mm-thick tape, able to transport electric current in the 100–1000 A range when cooled down to 77 K by liquid nitrogen (LN2). The REBCO superconductors have a great potential for use in many applications, such as power transmission cables, generators, motors, magnets or fault current limiters promising increased efficiency and reduced operating expenses [1].

The latter application is used for reducing the damages caused by short-circuit currents, known as fault currents, in highly interconnected power networks [2]. A resistive type of superconducting fault current limiter (SCFCL) utilizes a very fast transition of the REBCO from superconducting to resistive (non-superconducting) state (the so-called quench). When a fault occurs, electric current exceeds the critical value (*I_c_*) of the superconductor, impedance of the SCFCL increases and thus limits the fault current magnitude. In this state, the tape resistance is governed by resistance of adjacent metallic layers—substrate and protection layer [3]. Low magnitude of limited current is advantageous; therefore, metallic layers should be as thin as possible to keep overall electrical resistance of the composite tape high [4]. During the limitation event, the SCFCL experiences electric current and voltage simultaneously and thus a large amount of heat is generated. The conduction of the heat through the interface CC tape/LN2 is very poor, because LN2 evaporates due to a high temperature difference in the solid–liquid interface, and forms a vapour film with thermally insulating properties [5,6,7,8]. Roy et al. [9] showed by means of modelling of HTS tapes for resistive type of SCFCL, that the key factor in success to readily attain the thermal stability is the heat capacity instead of thermal conductivity. These magneto-thermal simulations reveal that increase in the heat capacity has a positive effect on thermal stability even if an increase in the heat capacity reduces the thermal diffusivity.

The length of CC tape needed in the resistive SCFCL is inversely proportional to the electric field, *E_max_*, which the tape withstands during current limitation. The SCFCLs designed with today’s commercially available CCs can be applied for electric fields up to 50 V m^−1^ [10]. A use of higher *E_max_* up to 150 V m^−1^ could considerably reduce the required superconductor length with a significant saving impact on overall costs. The ability of the CC tape to withstand higher electric field predominately depends on amount of generated heat, which can rapidly (typically in 50 ms) overheat the tape to more than 250 °C. Such high temperatures may permanently degrade the superconductor [11], and in the worst case, the HTS cannot retrieve its superconducting state after the fault. Therefore, the thermal stability of commercial CC tapes is an important issue. Several concepts are developing in frame of the Horizon 2020 project—FastGrid [12]. One of them is the so-called “high heat capacity concept”. This concept is based on heat absorption increase of the tape by adding of a layer with high heat capacity, *c_p_*.

Except water, high *c_p_* is a feature of many other materials, such as wood, soil, rubber, polymers and several metals (Li, Mg, Al); however, only a few materials meet other essential requirements for the thermal stabilization of the CC tapes. These requirements are related to thermal expansion, which should be as close as possible to thermal expansion of the CC tape, electrical resistivity, thermal conductivity, preservation of the tape flexibility, resistance against thermal shock as well as its properties must not significantly degrade in a cryogenic environment.

Conventional materials used to increase the heat capacity of the tape are brass, stainless steel or Hastelloy in compact [13,14,15] or porous form [16]. Drawback of these materials is their electrical conductivity, which reduces electrical resistivity of the tape and causes that more heat is generated at the same level of electric field [17]. Aluminium nitride ceramics is an electrical insulator with good thermal conductivity [18], but is difficult to prepare in form of a sufficiently thick layer. The thermal stabilization from a polymeric material like epoxy resin could be easily prepared by coating the CC tape with relatively low costs. Another advantage of such thermal stabilization is that it is electrical insulator, thus it does not lower the overall resistance of the conductor.

Epoxy resins like Stycast are known to be used in cryogenic applications [19], for instance, as an encapsulating material for HTS coils to prevent movement of CC tapes and to ensure even distribution of loads [20,21,22,23]. Generally speaking, coefficient of thermal expansion (CTE) of epoxy resins is much larger than CTE of Hastelloy C276^®^, usually used as a substrate of CC tapes, and their thermal conductivity is very low. As reported in [24,25,26], the mismatch in the thermal expansion could be reduced to reasonable value using an addition of filler material with very low (or even negative) CTE. This concept has simultaneously an advantage in enhancement of thermal conductivity [23,24,25,26,27,28,29].

We investigated the effect of various types of fillers on thermal properties of epoxy-based composites in our previous work [30]. Based on these experiments, a filler material with the best results was chosen and used for preparation of samples investigated in this work. The thermal stabilization layer was prepared from different epoxy resins mixed with the same filler at various filler/matrix ratios. The present paper studies the following properties of these composites: (1) thermophysical properties, (2) behaviour of the composites upon mechanical loading and thermal shock, and (3) the effectivity of thermal stabilization during limitation experiments.

## 2. Experimental

### 2.1. Materials

The samples were prepared from two component epoxy resins mixed with a SiC powder filler with various resin to SiC ratios. The selection of the SiC filler is based on the results described in [26]. The powder contains particles with of 200–450 mesh, as declared by the producer; we verified by scanning electron microscope that the particle diameter ranges from 3 µm to 100 µm. As the epoxy resins, Stycast 2850 FT and Araldite DBF were chosen because of good experience with them in LN2 environment. Furthermore, they possess excellent electrical insulation, proper adhesion properties and good resistance against thermal shock. A low value of viscosity was also taken in the account by their selection.

The Stycast 2850 FT was mixed with two different kinds of curing agents: Catalyst 11 or Catalyst 23LV. A disadvantage of the Catalyst 11 is a requirement of two-stage curing procedure at elevated temperature; on the other side, it has unusually low CTE from all selected resins. The Stycast hardened with the Catalyst 23LV as well as the Araldite DBF mixed with the hardener HY956 are both curable at room temperature (RT).

Scanning electron microscopy investigation revealed, that the Stycast, in contrary to the Araldite, contains in as delivered state already some amount of an Al_2_O_3_ filler. Its amount was estimated to be about 47 vol.% from several cross-sectioned samples. An example shown in Figure 1 shows evenly distributed Al_2_O_3_ and SiC ceramic particles without a hint of settling effect. Selected thermophysical properties of used pure materials and details about curing procedure are listed in Table 1.

### 2.2. Sample Preparation

Three kinds of samples were prepared for experiments we have utilized in systematic selection of the thermal stabilization layer:Bulk samples of dedicated shape for measurements of thermo-physical properties of high-*c_p_* materials,Hastelloy/Ag tapes coated with the high-*c_p_* layer for mechanical testing and thermal cycling,Superconducting tapes coated with the high-*c_p_* layer for limitation experiments.

In the preparation of composite materials for thermal stabilization, as the first step, the weighed quantities of the resin and catalyst were mixed thoroughly, then a defined amount of the SiC filler was added, followed by mechanical stirring of the mixture for 10 min at RT. The initial SiC content of 10 vol.% was based on several published works [23,27,28].

In order to obtain bulk samples for the measurements of thermophysical properties, cylindrical-shaped containers of required dimensions were filled with the prepared mixture and cured.

The samples for mechanical testing, thermal cycling and electrical measurements were prepared by coating of the paste-like composite on 12-mm-wide tapes manually by stainless razor blade with a help of leading rails with defined height, followed by the curing. In case of samples for testing of mechanical properties, the thicknesses of the layer measured by a micrometer screw gauge were in range from 120 µm to 500 µm with standard deviation not higher than 30 µm. The coatings were applied on the top as well as on both sides (top and bottom) of the tape. Other samples were prepared as top-coated with about the same layer thickness 175 ± 25 µm. Approximately 8 mm × 1 mm roving of E-type glass fibres with a fibre diameter of 14 µm was added into the coating for several samples to improve their mechanical properties. The fibres were encapsulated in the coating before the curing process by pushing the paste-like composite into the roving fixed at its ends.

As the CC tape, a 12-mm-wide superconducting tape from THEVA Company with *I_c_* about 600 A was used. The layer thicknesses of used CC tape are depicted in Figure 2a. The ends of the bare CC tapes subjected to limitation experiments were electroplated with additional copper layer approximately 10-µm-thick as shown in Figure 2b. The thermal stabilization layer was then coated with overlap of 3 mm to the Cu area. This modification step was necessary to avoid a damage of the sample due to insufficient thermal stabilization at the bare ends of the tapes.

A dummy 12-mm-wide Hastelloy tape with a 1.6-µm-thin Ag layer, deposited in the same company, was used instead of the CC tape in case of all samples subjected to mechanical testing and thermal cycling in order to save the material costs.

Table 2 gives an overview of all composites used for preparation of the investigated bulk samples and the samples of CC tapes with the thermal stabilization layer.

### 2.3. Experimental Procedures

The thermal expansion was measured on 12-mm-long cylinder bars with a 4.7 mm diameter using a Netzsch DIL 402C dilatometer (Netzsch, Selb, Germany) in helium environment with a heating rate of 3 °C min^−1^ over the temperature range from −140 to 200 °C. A correction measurement using certified alumina standard was done before each batch of samples to ensure the most accurate results of the coefficient of thermal expansion. The CTE was determined from the slope of the thermal expansion-temperature dependence in two temperature ranges: from −100 to 20 °C and from 130 to 200 °C because of glass transition temperature, *T_g_*.

The heat capacity was measured on a calibrated PerkinElmer Diamond differential scanning calorimeter (DSC) (Perkin Elmer, Waltham, MA, USA) in N_2_ atmosphere with a heating rate of 10 °C min^−1^ over the temperature range from −60 to 200 °C. The mass of the samples was varying about 35 mg. All samples were measured three times. The first DSC curve was slightly influenced by finishing of the polymerization process, the second and the third measurements were almost identical. The data from the last measurement were used for the determination of *T_g_*.

Measurements of the thermal diffusivity (*α*) were carried out using NETZSCH LFA 427 laser flash analyser (Netzsch, Selb, Germany). The cylindrical shaped samples (Ø 12.4 mm, 2 mm thick) were coated with graphite and measured at various temperatures below *T_g_* in static air atmosphere, using a laser voltage of 480 V and pulse width of 0.4 ms. The results were refined by Cape Lehman + pulse correction model. The thermal conductivity, *κ*, was calculated using the equation
*κ* = *ρ*·*α*·*c_p_*(1)
where *ρ* is the density of the sample, *α* is the thermal diffusivity and *c_p_* is the specific heat capacity. The Mettler Toledo precision laboratory scales and Archimedes method were used for determination of the density.

The flexibility of the thermal stabilization was proven by bending the CC tapes with one-sided and two-sided coatings of the selected composition and different thicknesses. The mechanical testing was performed at RT by fixing the samples at one end and bending it manually on a half-cylindrical former with various bending diameters from 350 mm to 10 mm. The winding of the samples was started from the highest bending diameter to smaller in steps until damage of the thermal stabilization layer was observed.

A thermal cycling of the CC tapes coated with selected thermal stabilization layer was carried out to test their resistance against thermal shocks. Firstly, 8-cm-long samples (without bending influence) were 300 times cooled down in LN2 bath for 30 s and repeatedly heated to 55 °C in an oven with air environment for 2 min. Longer samples with lengths of 30 cm were tested as bent on bending diameter of 25 cm. Its thermal cycling was carried out in temperature range from LN2 (dwell time 30 s) to RT (dwell time 3 min), then to 100 °C (dwell time 5 min) and finally to 150 °C (dwell time 5 min).

In the limitation experiments, the circuit schematically shown in Figure 3 was utilized. The source of direct current (DC) voltage, *U_app_*, was connected to the sample for the required time (50 ms), by the solid state power switch. The electric field and current were registered by data acquisition system with simultaneous sampling. The voltage level was controlled by the number of battery cells connected in series. From 6 to 9 cells were utilized to reach the electrical fields from 100 to 160 V m^−1^ on 11-cm-long samples. The values of circuit resistance and inductance were typically around 6 mΩ and 6 µH, respectively. Similar circuit but with less powerful supply was used for critical current measurement performed in order to check a possible tape damage during current limitation event.

An amount of heat, *Q*, produced in the sample during the limitation event was calculated as
(2)$$Q=\int_{0}^{t}IEdt,$$
where *I* is electric current, *E* is electric field and *t* is required time. The amount of the heat produced during the fixed pulse time was larger for higher applied voltages. The temperature of the sample was calculated from measured electrical resistance of the samples based on a calibration measurement.

## 3. Results and Discussion

The results were achieved in several steps. First, the thermophysical properties like thermal expansion, specific heat capacity, thermal diffusivity and thermal conductivity of different bulk composites were measured in order to find the most suitable candidate for use as thermal stabilization of CC tapes. Then selected composites were coated on CC tapes and their flexibility and resistance against thermal shocks was tested. Finally, electric current limitation experiments were performed to evaluate their thermal stabilization performance.

### 3.1. Thermal Expansion

The thermal expansion matching between the thermal capacity layer and the superconducting tape is of high importance in a modification of the tape, because too large difference would lead to delamination and cracking effects when the sample is bent and subjected to rapid temperature change.

The results of relative thermal expansions measured in the bulk samples are depicted in Figure 4. The thermal expansion of all investigated systems increases almost linearly with the temperature. After reaching the *T_g_*, the increasing trend continues, however, with a higher slope. These measurements demonstrated that the SiC filler has a positive impact on thermal expansion reduction of composites. The best results were achieved by mixing the resins with the highest amount of the SiC powder. The temperature behaviour of the thermal expansion was modified in the samples Ar-SiC40 (solid blue curve in Figure 4) and SB + 23LV-SiC20 (solid red curve in Figure 4) in a very similar way. The closest thermal expansion to the one of Hastelloy C276^®^ (black line in Figure 4) was observed for the sample SB + 11-SiC20 (solid green curve in Figure 4).

The CTE was determined from the slope of the thermal expansion-temperature dependence in two temperature ranges: from −100 to 20 °C and from 130 to 200 °C, because the glass transition temperature is in the range between 59 °C and 127 °C. The calculated CTEs for all measured systems are listed in Table 3. The lowest achieved CTE_RT_ belongs to the SB + 11-SiC20 composite. The addition of alumina with CTE_RT_ = 6 × 10^−6^ K^−1^ by producer reduces probably the CTE_RT_ of Stycast matrix (in comparison with the Araldite) significantly. Our addition of SiC filler with CTE_RT_ = 4 × 10^−6^ K^−1^ contributed to the further CTE_RT_ reduction of the composite. The CTE_RT_ = 17 × 10^−6^ K^−1^ is the closest value to the CTE_RT_ of Hastelloy C276^®^, which is 12 × 10^−6^ K^−1^.

### 3.2. Heat Capacity

This is the essential physical quantity in our high-*c_p_* concept, where the coating material should possess a high heat capacity. The resins as received from producer (hereinafter referred to as pure) listed in Table 1 meet lightly this condition; however, they need to be mixed with a component which has lower *c_p_*, and one could expect that such mixed composite will also decrease the specific heat capacity. The performed measurements confirmed the influence of the filler addition on the *c_p_* change by the aforementioned way. The decrease is quantified in Figure 5. The highest *c_p_* decrease was observed for the Ar-SiC40 sample due to 40 vol.% of SiC addition (drop about 0.6 J g^−1^ K^−1^), nevertheless the *c_p_* of the most doped Araldite composite has still a level of the both Stycast-based pure resins. In the SB + 23LV-SiC20 and SB + 11-SiC20 samples, the *c_p_* values 0.96 J g^−1^ K^−1^ and 0.88 J g^−1^ K^−1^ were measured at RT, respectively, and the decrease was approximately the same in both cases (about 0.1 J g^−1^ K^−1^) when compared with the pure resins. In general, the measurements ensured us that all studied composites keep the heat capacity still high enough for the intended purpose.

The heat capacity measurements were used for determination of *T_g_*, which are listed in Table 3. The presence of the glass transition is a double-edge issue. On one side, the *T_g_* itself is an endothermic event, and helps to absorb the heat produced during fault current limitation. On the other side, the thermal expansion of the composites at temperatures higher than the *T_g_* increases significantly, and it can cause delamination of thermal stabilization layer, and so impairment of its function. The effectivity of the composite layer will probably be dependent on how high is the temperature, which can be reached in the stabilizing coating during limitation event lasting 50 ms. Depending on applied electric field, the temperatures about 100 °C could be expected. From this point of view, the system SB + 11-SiC20 with the highest achieved *T_g_* seems to be the most appropriate for our purpose.

### 3.3. Thermal Diffusivity and Thermal Conductivity

The ability of material to conduct heat is also important issue regarding to the heat sinking. Generally, polymers possess a higher heat capacity, but lower thermal conductivity, because their atomic density is relatively low. However, the adding of filler with high thermal conductivity such as SiC (250 W m^−1^ K^−1^) in the low conductive epoxy matrix can improve the mentioned deficiency.

The thermal diffusivity was measured for the investigated samples and the values of thermal conductivity were calculated from obtained values of *α*, *c_p_*, and density. The densities of the investigated systems are listed in Table 3. The temperature dependencies of the thermal diffusivity and thermal conductivity are shown in Figure 6.

These plots confirmed that the thermal diffusivity and thus thermal conductivity can be enhanced by powder addition, obeying the rule: the higher the filler content, the more thermally conductive composite is. The SiC content of 40 vol.% increased the conductivity of Araldite resin about 10 times. Hence, the added SiC content was smaller in the Stycast-based resins; the thermal conductivity did not significantly improve (about 2.5 times for both). On the other hand, the Stycast 2850 FT contains in as delivered state already 47% of Al_2_O_3_ filler, therefore the thermal conductivities of pure Stycast resins are higher than that of the Araldite. The best thermal conductivity 3.1 W m^−1^ K^−1^ was achieved for the SB + 11-SiC20 sample.

The different size of filler particles plays significant role for achievement of a good thermal conductivity. During the mixing, the small particles fill the gaps between the bigger angular SiC plates, so the particles in sufficient amount can touch each other. By this mean, the particles create quasi-conductive paths for heat flow as suggested in [37].

### 3.4. Flexibility at Room Temperature

The superconducting tape, after coating with about 150–200 µm-thick thermal stabilization layer, should preserve a certain flexibility since the tape needs to be wound into a form of a pancake coil. Therefore, mechanical tests by bending the coated tapes were performed at room temperature (i.e., at temperature of coil manufacturing). Only the composites with the most appropriate thermophysical properties were selected: Ar-SiC40, SB + 23LV-SiC20 and SB + 11-SiC20. Based on our previous experiments, we know that the flexibility of the composite layer changes with its thickness. For this reason, dummy CC tapes (without superconducting layer) covered with the composite coatings of various thicknesses were prepared. In addition, two kinds of samples were tested: coated from one side (original concept) and coated from both sides. We suppose that the tape with double-sided coating could have several advantages. The doubling the amount of high-*c_p_* material helps to reduce more efficiently the maximum temperature reached during the quench. In addition, the mechanical forces due to thermal expansion/contraction should be less disturbing because of the symmetrical action on both sides of the tape.

In Figure 7, the summarized results showing how the bending diameter depends on the coating thickness. In the case of double-sided coatings, the thickness depicted in the plots is the mean value of coating thicknesses located at the top and bottom of the tape. The values in the graphs represent the state at which the sample was still not damaged, and after which the next reduction of bending diameter caused cracking or delamination of the tested coating. From Figure 7, it is obvious that the flexibility of the coatings improves with the decreasing of the coating thickness. All tested systems revealed a high bending ability when the thickness of the composite coating is below 200 µm (intended thickness for use in our next experiments). The Ar-SiC40 and SB + 23LV-SiC20 one-sided and double-sided coatings can be securely bent at RT on bending diameters down to 5 cm.

The SB + 11-SiC20 coating was a little bit stiffer after the curing procedure, but its bending possibilities are still excellent: in case of one-sided coating down to 10 cm bending diameter, and in case of double-sided coating down to 5 cm bending diameter. Considering the smallest bending diameters expected during pancake preparation foreseen in frame of the project [12], which is 25 cm, the tested composite materials for high-*c_p_* layer are flexible enough at room temperature.

### 3.5. Resistance Against Thermal Shock

The CC tape applied in SCFCL has to withstand rapid temperature change inducing a thermal shock during fault current events. Up to now, the behaviour of composites under thermal shock condition was investigated using only a simple test by immersion of short CC tapes covered with composite coating into the LN2 bath. The composites having insufficiently low CTE bent immediately the CC tape during such test. This effect was not observed in case of coatings Ar-SiC40, SB + 23LV-SiC20 and SB + 11-SiC20. Furthermore, their resistance against thermal shock was tested by thermal cycling on short samples with length of 8 cm. After 300 cycles with alternating the temperatures from −196 °C to 55 °C, no visible damage in form of cracking or delamination was observed.

These tests have encouraged us to check the same coatings on longer tapes (30 cm), which were bent to the required diameter of 25 cm simulating the pancake shape, and then subjected to thermal cycling. At such conditions, the risk of a delamination is significantly higher. Although the thermal cycling was chosen in a moderate temperature range from LN2 to RT, none of tested samples withstood this stress. As an example, Figure 8a shows how the coating was locally damaged by cracking followed by delamination already after several repetitions (up to 8 times). We suppose that this behavior is due to high stress at the interface of the CC tape/coating due to differences in CTE.

Therefore, we modified the SB + 11-SiC20 coating in two ways, which up to that point, had the lowest CTE. The first one was a reinforcement of the coating by inserting roving glass fibres into it before curing (sample SB + 11-SiC20-GF). The second idea is a further reduction of CTE by the increase of the filler content from 20 to 30 vol.% (sample SB + 11-SiC30). However, the ratio with 20 vol.% of SiC was a maximum amount in the SB + 11 system, which still allowed application of a simple coating procedure of the CC tape with a paste-like mixture. In order to insert a higher amount of filler into the resin, the mix ratio resin to hardener was adapted (see Table 2). Two new samples were then subjected to thermal cycling test under the conditions, at which the cracking was before observed. The samples SB + 11-SiC20-GF and SB + 11-SiC30 withstood 10 thermal cycles from LN2 to RT without any damage. The next thermal cycling from LN2 to 100 °C (20 times) and from LN2 to 150 °C (30 times) confirmed their extraordinary resistance against thermal shocks. Both samples after thermal cycling are shown in Figure 8b.

### 3.6. Fault Current Limitation

An effectiveness of the composite thermal stabilization was experimentally investigated by simulation of a fault current event in the samples during an experiment called limitation test. Mapping the performance of a reference bare CC tape without the high-*c_p_* layer, on which only the ends of the tape were electroplated by Cu layer as show in Figure 2b, was the starting point of these investigations. It was found that 10 limitation cycles with 4 battery cells, corresponding to electric field with the value 65 V m^−1^, did not change the critical current of the bare CC tape. After first limitation cycle with 5 cells (85 V m^−1^), 25% decrease from initial *I_c_* was observed, second and third cycle increased the *I_c_* degradation to 40% and 65% of its initial value, respectively. Two new bare CC tape samples were used for the next limitation experiment using 7 cells (*E* = 130 V m^−1^) and 9 cells (*E* = 150 V m^−1^). During both limitations, more than 85% of *I_c_* degradation was immediately recorded on these samples without additional high-*c_p_* layer.

The electric field achieved with 7 cells was proposed as starting point for the limitation test of CC tape samples covered by the SB + 11-SiC20-GF and SB + 11-SiC30 coatings, which showed good resistance against thermal shock during thermal cycling. The samples with the coatings SB + 11-SiC20 and Ar-SiC40 were added to the testing just for comparison. Figure 9a shows normalized values of critical current measured in the samples after each of 10 limitation cycles carried out with 7 cells. An evolution of the electric field and current during the limitation pulse is depicted in Figure 9b. After initial steep increase of the current, the sample loses superconducting properties and the electric current flows into metallic parts of the tape with following decrease of the current. The electric field was not the same for particular samples because its level depended on battery cells charging. For instance, the SB + 11-SiC20-GF sample sustained the lowest electric field (120 V m^−1^), while the SB + 11-SiC30 sample was subjected to the highest electric field (135 V m^−1^). Important is that the *I_c_* was not degraded in the tapes with high-*c_p_* layer, except a tiny degradation in the system SB + 11-SiC20. Surprisingly, the second sample with higher CTE_RT_ difference between coated and coating materials (Ar-SiC40), did not show any *I_c_* degradation.

The temperature behaviour of the samples during limitation with 7 cells is shown in Figure 9c. As evident in this plot, the temperatures reached in the coated samples during this limitation experiment are significantly lower (<55 °C) in comparison with the bare sample (170 °C). Since the measured temperatures probably did not reach *T_g_* of the coatings, no significant stress due to higher CTE above this temperature was generated, and the samples could show a good limitation performance. This is valid also for the critical samples (Ar-SiC40, SB + 11-SiC20) because the samples were short, and such samples showed no cracking during the previously performed thermal cycling.

In the next, all samples were subjected to the limitation testing with higher electric field, which was increased using 9 cells to about 150–160 V m^−1^ (see Figure 9e). An overview of samples behaviour at such high electric fields is visible in Figure 9d. The *I_c_* of the SB + 11-SiC20 samples started to decrease already after the 1st cycle, and this decrease was continuous with rising number of limitation cycles up to approximately 34% degradation of *I_c_* was reached. Similarly, the continuous *I_c_* decrease was also detected in the SB + 11-SiC30 sample, but with delayed start (after 5th cycle). It should be taken into account, that this sample sustained about 10 V m^−1^ higher electric field in comparison with the SB + 11-SiC20 sample, and eventually it could withstand more. The sample reinforced with glass fibers (SB + 11-SiC20-GF) was showing only small decrease in critical current (less than 5%), observed after 4th cycle. After 9th cycle, the *I_c_* suddenly dropped by about 40%. The best performance seems to be for the Ar-SiC40 sample, which *I_c_* decrease was about 6% even after 10th cycle. However, the coating was during limitation visually damaged by the cracking formation as shown in Figure 10. An investigation of the limited samples using a light microscopy revealed that, except the SB + 11-SiC20 sample, which coating was additionally delaminated, also the samples SB + 11-SiC20 and SB + 11-SiC30 became cracked in the coating. The SB + 10-SiC20-GF was not cracked.

The SB + 11-SiC30 and SB + 11-SiC20-GF composites did not show cracking during thermal cycling up to 150 °C when coated on longer CC tapes (see Figure 8b). This indicates that the temperature during the limitation with 9 cells could exceed this value. According to the plot in Figure 9f, the coated samples indeed suffer such high temperatures (155 °C for all Stycast-based samples, 184 °C for Ar-SiC40), what means a higher difference in CTE between the coating and the CC tape causing the crack formation. If the CC tape does not have the additional thermal stabilization layer, it is overheated to 215 °C.

Figure 11 shows the temperatures depicted as a function of Joule heat, which is in the sample generated during the quench. The better the thermal stabilization capability of the additional layer, the lower temperature will be measured and the higher amount of Joule heat could be generated in the sample, which is absorbed by suitable high-*c_p_* material. The best heat absorption ability can be attributed to the SB + 11-SiC30 sample in both cases of applied electric field. The reason for this behaviour is not fully supported by experimental data as of yet. As showed an additional cp measurement for this composite, the heat capacity was not reduced in comparison with the SB + 11-SiC20 sample. Therefore, we assume that for the higher heat absorption capability, the highest filler ratio in the Stycast matrix (30% of SiC + 46% Al_2_O_3_) is responsible. Due to formation of conductive paths for heat flow, the produced heat could be absorbed more effectively, that means also inside the composite coating.

The results of limitation experiments are promising. The coating of the CC tape with 200 µm thick epoxy-ceramic filler composite is worth doing because a significant improvement was achieved under the electric fields up to 130 V m^−1^, in comparison with the not coated CC tapes withstanding maximal electric field 65 V m^−1^. The reason for that is the ability of the coating to absorb the generated heat, which results in considerable decrease of the CC tape temperature (about 115 °C).

The best results showed the SB + 11-SiC30 and SB + 11-SiC20-GF composites with an excellent resistance against the thermal shock, and together with the Ar-SiC40 sample, they had the lowest *I_c_* degradation after limitation using electric field up to 160 V m^−1^. However, the expected correlation between the *I_c_* degradation and the appearance of the cracks formed during limitation with such high electric field has not been fully confirmed. When the coating is not reinforced for instance with glass fibers, high temperatures cause cracks in the high-*c_p_* layer due to difference in CTE. When the coating is reinforced, the mechanical properties of the coating are significantly improved, but finally an abrupt decrease of *I_c_* was also observed. We assume that it happens due to high strains at the coating/CC tape interface, when the superconducting layer was probably not able to withstand higher dilatation of reinforced coating and cracked. The formation of cracks in thermal stabilization layer need not be always responsible for *I_c_* degradation. The Araldite coating was cracked, but as indicated by the results in Figure 9d, it still protects the CC tape during its quench, although the temperature was rising. This is an interesting result, but it is important to know if the cracked coating will delaminate when coated on a CC tape with a length of several meters and the tape has a curled form. Such a development is to be expected, and was taken into account during the previous observation of the thermal cycling.

## 4. Conclusions

The possibility to increase the robustness of CC tapes has been studied, in terms of the electric field they can sustain during the fault current limiting event, by adding a layer with high heat capacity on the tape surface.

The additional layer was prepared from composite material, by mixing the SiC powder with epoxy resin cured with suitable hardener. In its composition, we varied the type of matrix, the content of filler particles and the use of reinforcement, and by this way, we modified the thermal properties of composites. Very promising results have been achieved on layers based on commercial resins like Stycast, in particular when reinforced by glass fibres. The CC tapes coated with up to a 200-µm-thick composite layer remain flexible and resistant against thermal shock, which were induced by temperature change from LN2 to 150 °C.

Considering the effectiveness of the thermal stabilization, the limitation experiments on short samples demonstrated sufficient reduction of the maximum temperature achieved during the quench with up to 130 V m^−1^ intensity of electric field. This is a significant improvement compared to the bare CC tape (65 V m^−1^). The achieved results of limitation experiments under higher electric fields (about 150 V m^−1^) are disputable, as there is no strong correlation between the presence of visible cracks in the coating and the degradation of *I_c_.*

## Figures and Tables

**Figure 1 materials-13-01832-f001:**
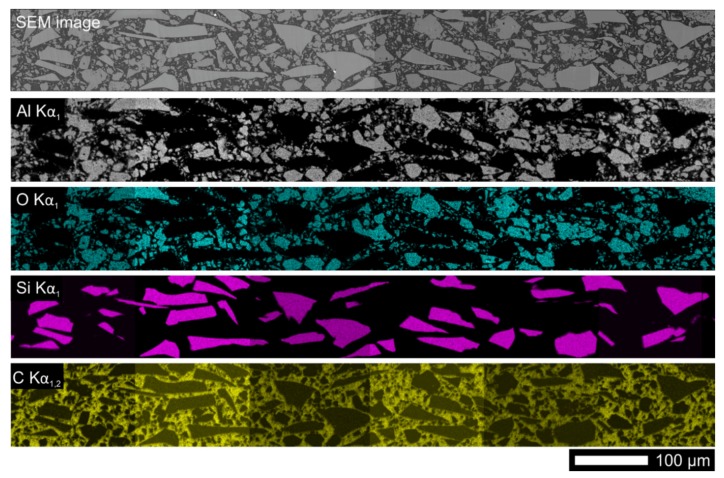
Distribution of ceramic particles in composite of Stycast 2850 FT resin (SB + 11) with 20% SiC given by mapping of Al, O, Si and C using X-ray energy-dispersive analysis.

**Figure 2 materials-13-01832-f002:**
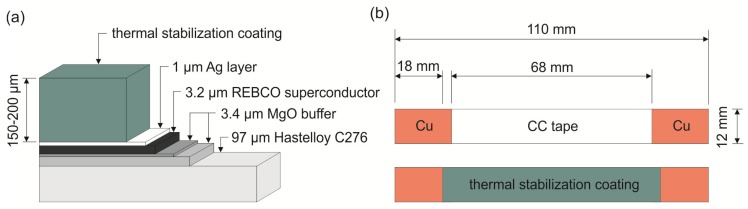
Coated coated (CC) tape samples with coated high-*c_p_* layer: (**a**) The layer thicknesses; (**b**) Dimensions of the samples for limitation experiments.

**Figure 3 materials-13-01832-f003:**
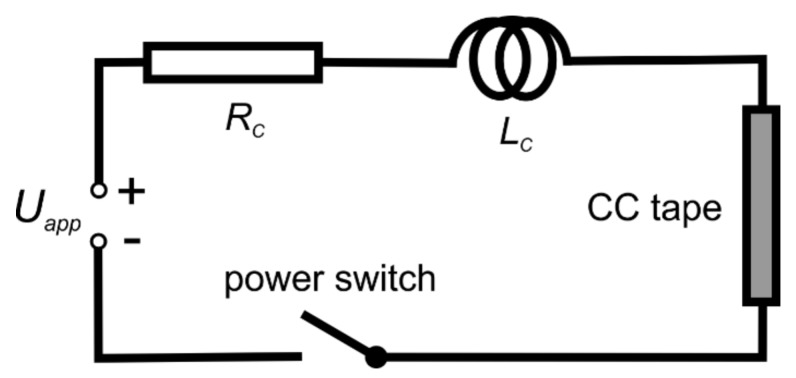
Schematic drawing of the circuit used in current limitation experiments. Set of accumulator battery cells serves as the power supply with voltage *U_app_*. *R_c_* and *L_c_* represent the circuit resistance and inductance, respectively. The signal wires allowing measuring the voltages on different portions of the circuit are not shown.

**Figure 4 materials-13-01832-f004:**
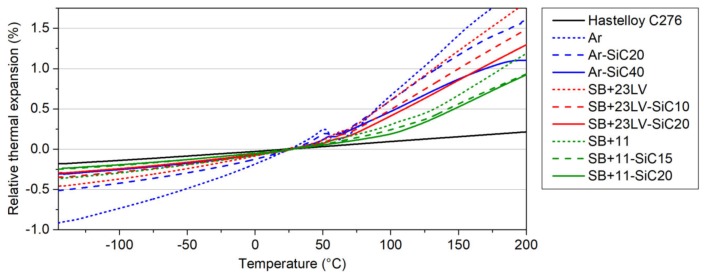
Temperature dependences of relative thermal expansion measured in the resins as received from producer and resin–SiC composites compared with a Hastelloy C276^®^.

**Figure 5 materials-13-01832-f005:**
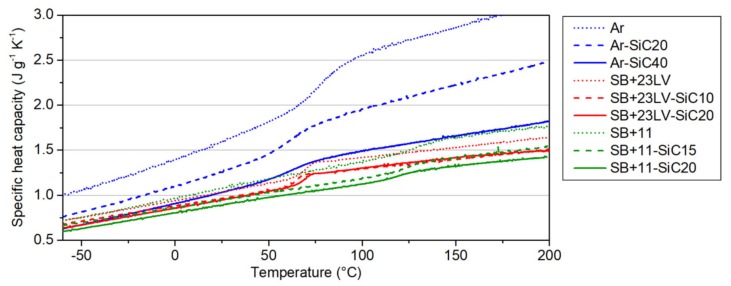
Specific heat capacities as a function of temperature measured in the pure resins and resin–SiC composites.

**Figure 6 materials-13-01832-f006:**
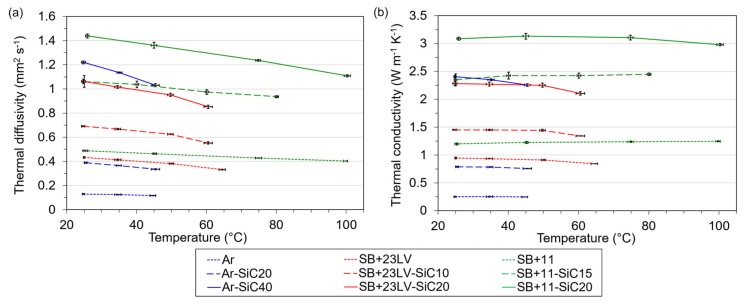
Temperature dependence of thermal diffusivity (**a**) and thermal conductivity (**b**) for prepared pure resins and resin–SiC composites.

**Figure 7 materials-13-01832-f007:**
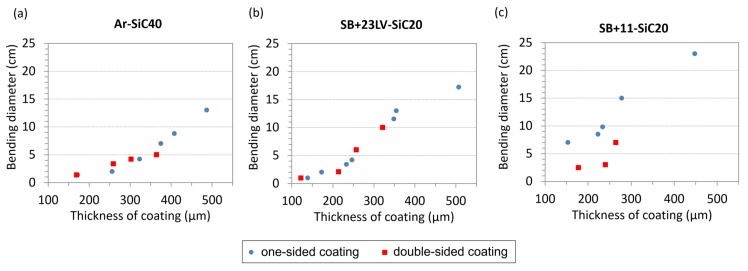
Dependence of minimal bending diameter, at which no damage was observed, on coating thickness found out by bending tests of Hastelloy/Ag tapes with (**a**) Ar-SiC20, (**b**) SB + 23LV-SiC20, and (**c**) SB + 11-SiC20 coating applied on one (blue circles) or both sides (red squares) of the tape.

**Figure 8 materials-13-01832-f008:**
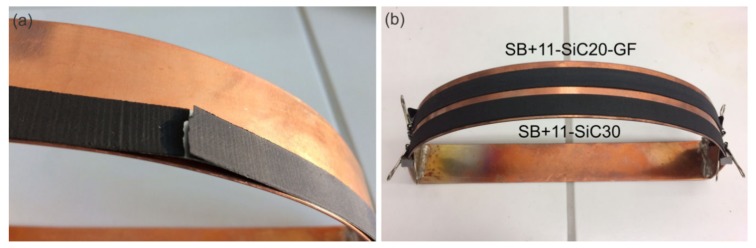
Results of resistance against thermal shock tests performed on 30-cm-long Hastelloy/Ag tapes covered with different thermal stabilization layers: (**a**) SB + 11-SiC20 cracked and delaminated due to thermal cycling from LN2 to RT; (**b**) SB + 11-SiC20-GF and SB + 11-SiC30 after thermal cycling from LN2 to 150 °C without visible damage.

**Figure 9 materials-13-01832-f009:**
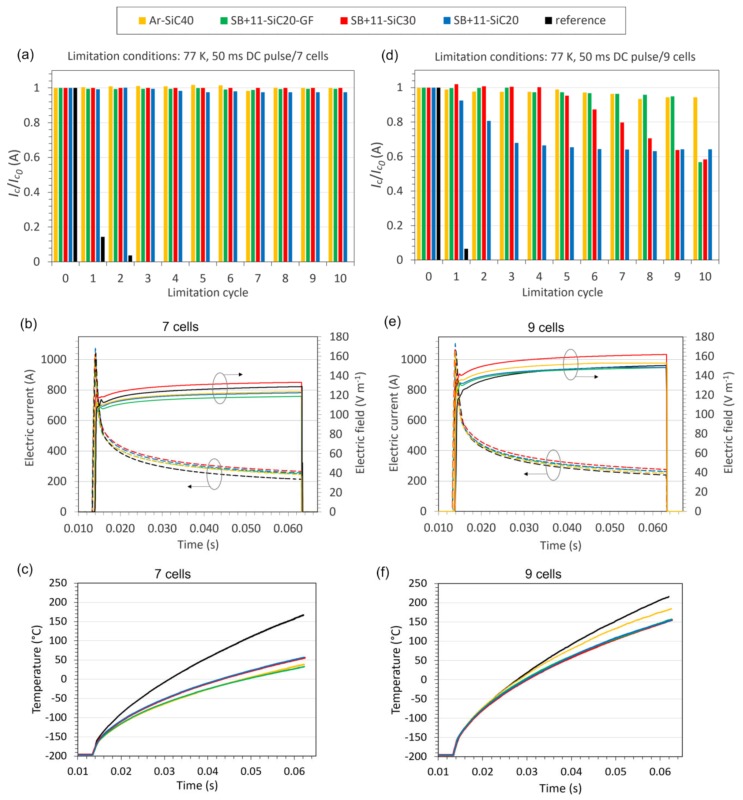
Dependence of normalized critical current on number of limitation cycle for investigated systems Ar-SiC40 (yellow), SB + 11-SiC20 (blue), SB + 11-SiC30 (red), SB + 11-SiC20-GF (green) and bare CC tape as reference (black) during limitation experiments with 7 (**a**) and 9 cells (**d**). Measured electric fields and currents are shown in (**b**) and (**e**). Temperatures developed in the samples using 7 and 9 cells are depicted in (**c**) and (**f**), respectively.

**Figure 10 materials-13-01832-f010:**
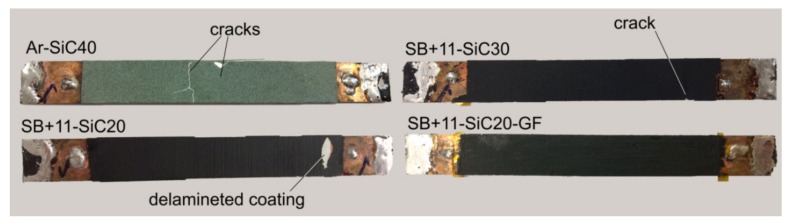
Samples after limitation experiment with 9 cells and 50 ms DC pulse.

**Figure 11 materials-13-01832-f011:**
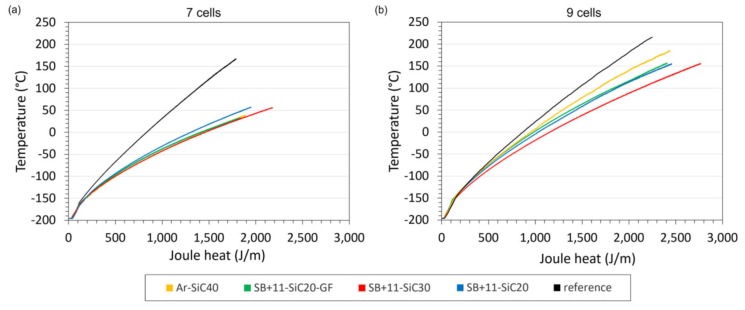
Dependence of temperature on Joule heat generated in the CC tapes during limitation experiment with 50 ms DC pulse and different electric fields 7 (**a**) and 9 cells (**b**).

**Table 1 materials-13-01832-t001:** Selected properties of used materials for preparation of samples.

Name	Material	Curing	*c_p_* at RT [J g^−1^ K^−1^]	*α* at RT [W m^−1^ K^−1^]	CTE_RT_ [× 10^−6^ K^−1^]
Ar	Araldite DBF + HY 951	48 h @ RT	1.60	0.25	78
SB + 23LV	Stycast 2850 FT + Catalyst 23LV	24 h @ RT	1.04	0.90	38
SB + 11	Stycast FT 2850 + Catalyst 11	1 h @ 125 °C + 4 h @ 150 °C	1.07	1.19	28
Al_2_O_3_	aluminium oxide	—	0.78 [31]	28 [32]	6 [32]
SiC	silicon carbide	—	0.75 [33]	250 [34]	4 [35]
GF	E-glass fibres	—	0.80 [36]	1.3 [36]	5 [36]

**Table 2 materials-13-01832-t002:** Overview of all composites used for preparation of bulk samples and coated coated conductor (CC) tapes.

Sample	Resin + Hardener	Mix Ratio	Filler
Ar	Araldite DBF + Aradur HY 951	100:10	—
Ar-SiC20	Araldite DBF + Aradur HY 951	100:10	20 vol.% of SiC
Ar-SiC40	Araldite DBF + Aradur HY 951	100:10	40 vol.% of SiC
SB + 23LV	Stycast 2850 FT + Catalyst 23LV	100:7.5	—
SB + 23LV-SiC10	Stycast 2850 FT + Catalyst 23LV	100:7.5	10 vol.% of SiC
SB + 23LV-SiC20	Stycast 2850 FT + Catalyst 23LV	100:7.5	20 vol.% of SiC
SB + 11	Stycast 2850 FT + Catalyst 11	100:8.5	—
SB + 11-SiC15	Stycast 2850 FT + Catalyst 11	100:8.5	15 vol.% of SiC
SB + 11-SiC20	Stycast 2850 FT + Catalyst 11	100:8.5	20 vol.% of SiC
SB + 11-SiC20-GF	Stycast 2850 FT + Catalyst 11	100:8.5	20 vol.% of SiC, reinforced with glass fibres
SB + 11-SiC30	Stycast 2850 FT + Catalyst 11	100:12	30 vol.% of SiC

**Table 3 materials-13-01832-t003:** Densities, *T_g_*, and CTEs of investigated samples at RT and their average values in two temperature ranges below and above *T_g_*.

Sample	Density (g cm^−3^)	*T_g_* (°C)	CTE_RT_ (×10^−6^ K^−1^)	Average CTE Below/Above *T_g_* (×10^−6^ K^−1^)
Ar	1.180 ± 0.006	59	78	58/122
Ar-SiC20	1.613 ± 0.001	57	54	32/93
Ar-SiC40	1.906 ± 0.003	59	32	20/63
SB + 23LV	2.104 ± 0.006	67	38	30/121
SB + 23LV-SiC10	2.104 ± 0.009	66	26	21/99
SB + 23LV-SiC20	2.271 ± 0.002	67	24	18/89
SB + 11	2.271 ± 0.007	127	28	22/100
SB + 11-SiC15	2.345 ± 0.001	122	17	16/76
SB + 11-SiC20	2.691 ± 0.004	122	17	14/78

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
