# Peer review of "Composite Heat Sink Material for Superconducting Tape in Fault Current Limiter Applications"

_materials, 2020, doi:10.3390/ma13081832_

Round 1
Reviewer 1 Report
Dear Authors,
Your article concerns the investigations of properties of composites material intended for superconducting tape. The article is characterized by good structure and composition, and its content is assessed as interesting and worth publishing. While reading the article, I had a few comments that, hopefully, will help shape the final version of your article.
line 95: "The powder contains particles with size of 200-450 mesh." - can you give the dimension in um?
line 101 and further: "low thermal expansion". Add: "coefficient"
line 105 and earlier: "contains in as delivered state already some amount of an Al2O3 filler. Its amount was estimated to be about 47 vol.%." - how was the amount of Al2O3 measured?
References 27-32 in Table 1 are citted with errors.
Is there a phenomenon of increasing the hardness of samples due to testing at elevated temperature?
Line 121 – repeated (doubled) information from line 107.
Figure 2: change "swich" to "switch".
Lines 198 and 199: check grammar.
Figure 5. The lines on the plots are not clear.
What type of bending was used: 3 points, 4 points?
line 304 - "From Figure 6 it is obvious that the flexibility of the samples improves with the decreasing of the coating thickness." - it is not clear to me how this can be determined. Rather, it is not the flexibility of the sample but the coating?
Discussion of the results is carried out without comparing them with information from the literature.
Table 3 is inserted before being referenced in the text.
References: format according to the journal’s guidelines. Most of the cited papers have been published quite a long time ago. I suggest citing several recent (last 2 years) articles from literature, especially MDPi Publishing House.
Correct the abbreviation of the journal name [23] and [26].
Author Response
Dear Referee,
thank you very much for the detailed review of my manuscript. Please find below the point-to-point response to your comments. Changes according the reviewer comments and additional small changes made to make the statements clearer are marked red in the manuscript, which you can see in the attachment.
Best regards,
Marcela Pekarcikova
1st reviewer:
- line 95: "The powder contains particles with size of 200-450 mesh." - can you give the dimension in um?
We measured the size of SiC particles by SEM. Their size was in wide range from 3 µm to 100 µm. We added this in the text. The mesh size is information from producer and it can be useful to know this.
- line 101 and further: "low thermal expansion". Add: "coefficient"
We added the “coefficient of” in the text on this place and further.
- line 105 and earlier: "contains in as delivered state already some amount of an Al2O3 filler. Its amount was estimated to be about 47 vol.%." - how was the amount of Al2O3 measured?
We performed measurements of areas rich on Al and O in cross-sectioned samples according to EDX maps. One example of such EDX maps we also added in the manuscript as Fig. 1. The volume percentage of Al2O3 was estimated as a percentage ratio of the total measured area taking into account the Cavalieri’s principle. The next figures were subsequently renumbered.
- References 27-32 in Table 1 are citted with errors.
It probably happened by conversion to PDF file due to linked references. We removed the linking. It should be now OK.
- Is there a phenomenon of increasing the hardness of samples due to testing at elevated temperature?
We did not measure the hardness of the coatings, but we suppose it is not the case. After the curing procedure, the hardening process was almost closed, and we did not observe any obvious changes after testing at elevated temperatures.
- Line 121 – repeated (doubled) information from line 107.
The repeated information was deleted.
- Figure 2: change "swich" to "switch".
We corrected the figure.
- Lines 198 and 199: check grammar.
We tried to correct the grammar.
- Figure 5. The lines on the plots are not clear.
The Fig. 5 was improved.
- What type of bending was used: 3 points, 4 points?
These tests reveal the elastic modulus of bending, flexural stress or flexural strain of a material. We used neither 3 points nor 4 points. It would be useful to known all these mechanical characteristics, but such equipment was not available for our measurement. In the first approach, we needed to know if the coatings are cracking during bending to certain bending diameter. Therefore, the mechanical testing was performed by very simple test; the tapes were fixed at one end and bent manually on a half-cylindrical former, as shown on Fig. 8(b). We tried to explain it more clearly in the text (page 5 line 176-177).
- line 304 - "From Figure 6 it is obvious that the flexibility of the samples improves with the decreasing of the coating thickness." - it is not clear to me how this can be determined. Rather, it is not the flexibility of the sample but the coating?
You are right. We replaced the expression “samples” with the expression “coatings” in the sentence.
- Discussion of the results is carried out without comparing them with information from the literature.
Comparison of our results with literature should be, indeed, an essential part of the “Results and discussion” section. However, to the best of our knowledge, we are not aware of any publication dealing with coating of sufficiently similar composition. A brief mention of different approaches of metallic shunt layer is in introduction (line 69-72). To compare the metallic and the electrically non-conductive thermal stabilization layer it is possible when the same testing conditions are applied. This is the aim of our research activities in the next future.
- Table 3 is inserted before being referenced in the text.
The table was relocated after mentioning it in the text.
- References: format according to the journal’s guidelines. Most of the cited papers have been published quite a long time ago. I suggest citing several recent (last 2 years) articles from literature, especially MDPi Publishing House.
We tried to use the latest knowledge from literature. Yes, there are several older references in our manuscript, especially references to thermo-physical properties of known materials. We have added some newer references, and renumbered the following references.
- Correct the abbreviation of the journal name [23] and [26].
Yes, “J“ was missing in the first abbreviation.
https://www.library.caltech.edu/journal-title-abbreviations
According to this source, the second one of the abbreviations should be correct; nevertheless, if the reviewer has a more official list, I would kindly ask him to send it to me.

Reviewer 2 Report
Manuscript: “Composite heat sink material for superconducting tape in fault current limiter applications”
Authors: Marcela Pekarčíková et al.
The authors present a study on manufacture and testing of coating tape composites. The manufacture, testing methods and obtained results are very interesting, hence, I think the study can gain attentions from readers. However, I have some comments/questions for the authors as following:
Table 1: There occurs error in the hyperlink that connects to the reference papers.
Page 5, Lines 174-179: The thermal cycling test method was introduced. However, I think It is not clear how the test was implemented. Is there any special equipment that was used to move the tested specimens repeatedly from LN2 bath, to RT and then heating environment (100 and 150 oC) for a certain dwell period? Considering that there was 300 times for each test, it seems not easy to do all manually.
Page 9, Figure 6 presents the bending dimeter vs thickness of coating. However, it is not clear how the bending test was conducted.
Are there any standards that guide the experiment tests?
The vol. % of SiC was selected from 15~40% as shown in Table 2. Is there any theoretical background or reference for this selection?
Table 3 shows that the CTE at RT and Tg performance tend to be better with increasing filler loading. The SB+11+SiC30 was selected as it is the most appropriate one claimed by the authors. However, why do the authors not show the results for SB+11+SiC30. Is it better than SB+11+SiC20 or SB+11+SiC15?
Page 7, line 229: “The lowest achieved CTERT belongs to the SB+11-SiC20 composite”. It is obvious from Table 3 that not only SB+11-SiC20, but SB+11-SiC15 also offers lowest CTE at RT.
Author Response
Dear Referee,
thank you very much for the detailed review of my manuscript. Please find below the point-to-point response to your comments. Changes according the reviewer comments and additional small changes made to make the statements clearer are marked red in the manuscript, which you can see in the attachment.
Best regards,
Marcela Pekarcikova
2nd reviewer:
- Table 1: There occurs error in the hyperlink that connects to the reference papers.
It probably happened by conversion to PDF file due to linked references. We removed the linking. It should be now OK.
- Page 5, Lines 174-179: The thermal cycling test method was introduced. However, I think It is not clear how the test was implemented. Is there any special equipment that was used to move the tested specimens repeatedly from LN2 bath, to RT and then heating environment (100 and 150 oC) for a certain dwell period? Considering that there was 300 times for each test, it seems not easy to do all manually.
It sounds crazy, but it was done manually in frame of a diploma thesis. The tested samples were moved from LN2 bath to oven with defined temperature without special equipment. The heating environment was air (added on page 5 line 183), because we did not expect any troubles with oxidation. The heating times in oven were long enough to ensure the reaching of required temperature (5 min). If the sample is able to withstand the thermal load during performed tests, it is a good assumption for use during limitation event.
- Page 9, Figure 6 presents the bending dimeter vs thickness of coating. However, it is not clear how the bending test was conducted.
- Are there any standards that guide the experiment tests?
No, we did not use standard bending tests. In the first approach, we needed to know if the coatings are cracking during bending to certain bending diameter. Therefore, the mechanical testing was performed by very simple test; the tapes were fixed at one end and bent manually on a half-cylindrical former, as shown on Fig. 8(b). We tried to explain it more clearly in the text (page 5 line 176-177).
- The vol. % of SiC was selected from 15~40% as shown in Table 2. Is there any theoretical background or reference for this selection?
We started with 10 vol. % SiC. It was based on several published papers [19, 23, 24]. The references for selection of initial SiC content we added to the text (page 4 line 123-124). Then we increased the filler content to maximal possible value, which allows still to produce a paste-like mixture necessary for application of a simple coating procedure.
- Table 3 shows that the CTE at RT and Tg performance tend to be better with increasing filler loading. The SB+11+SiC30 was selected as it is the most appropriate one claimed by the authors. However, why do the
authors not show the results for SB+11+SiC30. Is it better than SB+11+SiC20 or SB+11+SiC15?
We measured the thermal expansion additionally also for SB+11-SiC30 and it was almost the same as for SB+11-SiC20. We have to say that all thermal expansion measurements were performed for bulk samples, but we use composite material in form of thin coating on the CC tapes. The latter discrepancy can significantly impact the CTE, otherwise we would not notice such big difference between these composites in form of thin coatings during thermal cycling. We plan to investigate the thermal expansion of coatings more in detail.
- Page 7, line 229: “The lowest achieved CTERT belongs to the SB+11-SiC20 composite”. It is obvious from Table 3 that not only SB+11-SiC20, but SB+11-SiC15 also offers lowest CTE at RT.
Yes, you are right. The difference is really tiny, in particular at temperatures below Tg. This is probably also reason that SB+11-SiC30 was not more improved. Above Tg, the difference is more visible and in favour for SB+11-SiC20, therefore we decided to test this one.

Reviewer 3 Report
The submitted manuscript entitled ‘Composite heat sink material for superconducting tape in fault current limiter applications’ deals with the production and testing (mechanical, electrical and thermal) of a layered, super-conductor based composite to improve their properties in fault limiting applications. The manuscript is basically well-written and sound, during its review, the following minor issues arose.
- Please solve every abbreviation at its first occurrence, even if they are well known and even if they are in the Abstract.
- In table 1, references are missing.
- ‘…as the first step, the weighed quantities of the resin and catalyst were mixed thoroughly, then a defined amount of the SiC filler was added, followed by mechanical stirring of the mixture for 10 min at RT.’ – how was the proper mixing of the SiC particles in the viscous epoxy checked? Was the settling of the particles avoided (checked) somehow?
- ‘In case of samples for testing of mechanical properties, the thicknesses of the layer were in range from 120 µm to 500 µm…’ – how was it measured? What about the evenness of the thickness along the samples?
- ‘Approximately 8 mm × 1 mm roving of E-type glass fibres with a fibre diameter of 14 µm was added into the coating for several samples to improve their mechanical properties. The fibres of the strand were evenly spaced in the coating before the curing process.’ – please detail how were the fibers placed into the material, how was the even distribution guaranteed and checked?
Author Response
Dear Referee,
thank you very much for the detailed review of my manuscript. Please find below the point-to-point response to your comments. Changes according the reviewer comments and additional small changes made to make the statements clearer are marked red in the manuscript, which you can see in the attachment.
Best regards,
Marcela Pekarcikova
3th reviewer:
- Please solve every abbreviation at its first occurrence, even if they are well known and even if they are in the Abstract.
We replaced abbreviation “CC” from the abstract by “superconducting tape”. In the next, the abbreviations are solved at its first occurrence as you recommended.
- In table 1, references are missing.
It probably happened by conversion to PDF file due to linked references. We removed the linking. It should be now OK.
- ‘…as the first step, the weighed quantities of the resin and catalyst were mixed thoroughly, then a defined amount of the SiC filler was added, followed by mechanical stirring of the mixture for 10 min at RT.’ – how was the proper mixing of the SiC particles in the viscous epoxy checked? Was the settling of the particles avoided (checked) somehow?
We checked this in cross-sectioned samples using SEM and EDX analysis. One example of such EDX analysis we added to the manuscript as Fig. 1. The ceramic particles were mixed and evenly distributed without a hint of settling effect. We mentioned this on page 3. The next figures were subsequently renumbered.
- ‘In case of samples for testing of mechanical properties, the thicknesses of the layer were in range from 120 µm to 500 µm…’ – how was it measured? What about the evenness of the thickness along the samples?
We performed a minimum of 15 thickness measurements of each sample along its length with a micrometer screw gauge. The thickness of the coating was calculated by subtracting of the bare CC thickness from overall thickness of the sample. In case of both-sided coated samples, the thickness depicted in Fig. 6 is the mean value of coating thicknesses located at the top and bottom of the tape (mentioned on page 9). The evenness of the thicknesses along the samples was of course not perfect. The standard deviation was mostly ± 20 µm but not higher than 30 µm. We added the STDEV value and the measuring tool in the text (page 4 line 129-130).
- ‘Approximately 8 mm × 1 mm roving of E-type glass fibres with a fibre diameter of 14 µm was added into the coating for several samples to improve their mechanical properties. The fibres of the strand were evenly spaced in the coating before the curing process.’ – please detail how were the fibers placed into the material, how was the even distribution guaranteed and checked?
The roving of glass fibres was spread manually on the CC tape and fixed at their ends. The composite was then applied on the top. With a little pressure we achieved that the paste-like consistence of the composite encapsulated the fibres. The distribution was probably not even and we did not check it. However, the main purpose was at first to see some effect of reinforcement. In the text, we corrected the sentence with not mentioning “evenly distributed” and tried to add more details about the placement of fibres into the coating material.

Round 2
Reviewer 1 Report
Dear Authors,
Thank you very much for all the answers to my comments and questions. I believe that the article can be published in its current form. Abbreviations for journalist names can also be checked at: https://www.letpub.com/index.php?journalid=3739&page=journalapp&view=detail
By the way, I want to congratulate graduate students so strongly involved (crazy ...) in scientific work!
Good luck!